# The structural basis of divalent cation block in a tetrameric prokaryotic sodium channel

Katsumasa Irie [1,2] ✉, Yoshinori Oda[3], Takashi Sumikama [4,5], Atsunori Oshima [2,3,6,7] & Yoshinori Fujiyoshi [8,9]

Divalent cation block is observed in various tetrameric ion channels. For blocking, a divalent cation is thought to bind in the ion pathway of the channel, but such block has not yet been directly observed. So, the behaviour of these blocking divalent cations remains still uncertain. Here, we elucidated the mechanism of the divalent cation block by reproducing the blocking effect into NavAb, a well-studied tetrameric sodium channel. Our crystal structures of NavAb mutants show that the mutations increasing the hydrophilicity of the inner vestibule of the pore domain enable a divalent cation to stack on the ion pathway. Furthermore, non-equilibrium molecular dynamics simulation showed that the stacking calcium ion repel sodium ion at the bottom of the selectivity filter. These results suggest the primary process of the divalent cation block mechanism in tetrameric cation channels.

Neural activity controls critical biological responses, such as thinking, memory formation, and muscle contraction. The transmission of neural activity occurs through various cation-selective channels on the nerve cell membrane[1]. Divalent cation block is observed in important tetrameric cation channels. For example, magnesium ions block the NMDA receptor's (NMDAR) current in a voltage-dependent manner[2]. Calcium ions also block the current of voltage-dependent calcium channels and IP3 receptors, known as calcium facilitation[3]. It has long been said that magnesium ions block the ion pathway of NMDAR[4,5].

The ion permeation pathway belongs to the pore domain. That of the tetrameric cation channels consist of two transmembrane helices, and the selectivity filter (SF) locates in the loop between them. The loop also contains one or two pore helices (P1 and P2 helix), and the SF follows the P1 helix (Fig. 1a). As the name indicates, the SF of the tetrameric cation channels is responsible for the selective permeation of cations. It also concerns the activity control of the channels, such as the C-type inactivation[6]. For NMDAR, the previous mutational analysis revealed that the residues at the SF (Asn616 of GluN1 subunit, Asn615,

Asn616, and Val618 of GluN2B subunit) are involved in magnesium inhibition[7]. A simulation study suggested the coordination structure of these residues to the magnesium ion[8]. Although recent studies revealed the structures of these receptors in various states[9–11], there are still no direct observations of the blocking magnesium ion. The blocking magnesium ion may be observed if the proposed coordination is stably formed. Therefore, it is possible that the blocking magnesium ion does not remain stable position in the channel but moves dynamically in the ion pathway. Because of the above situation, we thought it would also be helpful if similar phenomena could be investigated in detail, even in other channels.

Prokaryotic voltage-gated sodium channels (BacNavs) were cloned from bacteria living in various environments and characterized[12–14], which provided many insights into the molecular basis of tetrameric cation channels[15–17]. We recently found that one homolog of an ancestor-like channel group of BacNav (AnclNav), named NavPp, had similar divalent cation block[18]. However, BacNavs did not show divalent cation block. Therefore, reproducing the divalent cation block on BacNavs and

[1]Department of Biophysical Chemistry, School of Pharmaceutical Sciences, Wakayama Medical University, 25-1, Shichibancho, Wakayama 640-8156, Japan. [2]Cellular and Structural Physiology Institute (CeSPI), Nagoya University, Furo-cho, Chikusa, Nagoya 464-8601, Japan. [3]Department of Basic Medicinal Sciences, Graduate School of Pharmaceutical Sciences, Nagoya University, Furo-cho, Chikusa, Nagoya 464-8601, Japan. [4]PRESTO, JST, Kawaguchi 332-0012, Japan. [5]Nano Life Science Institute (WPI-NanoLSI), Kanazawa University, Kanazawa 920-1192, Japan. [6]Institute for Glyco-core Research (iGCORE), Nagoya University, Furo-cho, Chikusa-ku, Nagoya 464-8601, Japan. [7]Center for One Medicine Innovative Translational Research, Gifu University Institute for Advanced Study, Gifu 501-11193, Japan. [8]Cellular and Structural Physiology Laboratory (CeSPL), Advanced Research Institute, Tokyo Medical and Dental University, 1-5-45, Yushima, Bunkyo, Tokyo 113-8510, Japan. [9]CeSPIA Inc., 2-1-1, Otemachi, Chiyoda, Tokyo 100-0004, Japan. ✉ e-mail: kirie@wakayama-med.ac.jp

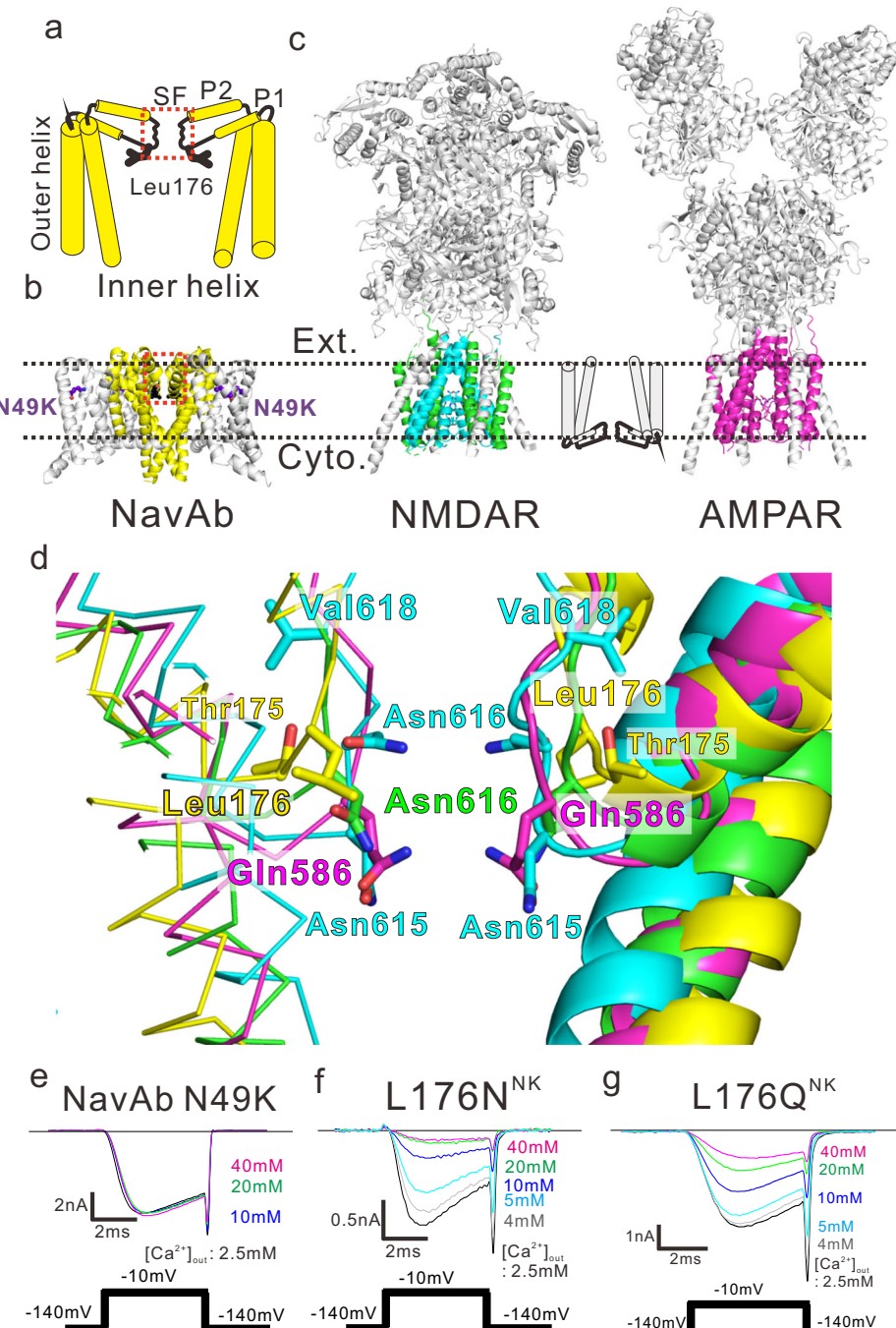

**Fig. 1 | Structural comparison among the pore domains of NavAb and NMDA and AMPA receptors. a** Schematic diagram of the pore domain of tetrameric cation channel. The front and rear subunits of the transmembrane part were removed for clarity. The pore domain contains two transmembrane helices (outer and inner helix). The Red dashed square indicates the selectivity filter (SF) at the centre of the pore domain. A black line depicts the position of Leu176 of NavAb. **b** Overall structure of NavAb N49K mutant (pdb code; 5yuc). The pore domain is coloured yellow. The Red dashed square indicates the selectivity filter (SF) at the centre of the pore domain. Leu176 is coloured black. N49K is coloured purple. **c** Overall structures of NMDAR and AMPAR. The pore domain of NMDAR (pdb code;

4tlm) is coloured green (NR1) and cyan (NR2B), and the pore domain of AMPAR (pdb code; 5wek) is coloured magenta. Ext. and Cyto. indicate extracellular and cytoplasmic sides, respectively. **d** The pore domain of NavAb superimposed on that of NMDAR NR1 and NR2b subunit and AMPAR A2 subunit, respectively. Pore domains were aligned with the P1 helix, the selectivity filter and the P2 helix (NavAb) and the pore helix and the selectivity filter (NMDAR and AMPAR). The NavAb, the NMDAR NR1, NR2b, and AMPA A2 subunits are coloured yellow, green, cyan, and magenta, respectively. **e–g** Representative current traces of NavAb N49K, L176N[NK], and L176Q[NK] mutants generated by −10 mV stimulation pulses in indicated extra-cellular Ca[2+] concentration solutions.

its structural analysis would help to elucidate their molecular mechanisms. Among BacNavs, NavAb, cloned from *Arcobacter butzleri*, is the most critical contributor, since its full-length structure of Nav at atomic resolution was first resolved[19,20]. It is a helpful template for understanding disease mechanisms, toxin activity, and human drug discovery targets[21–23].

In this work, we successfully reproduce the divalent cation-blocking effect on NavAb by introducing single mutations into the SF. Using these mutated NavAb, we conduct structural analysis and non-equilibrium molecular dynamics (MD) simulation to elucidate the process of molecular mechanisms of divalent cation-blocking.

## Results and Discussion

We superimposed the diagonal subunits of NavAb onto the diagonal NMDAR NR1 subunits aligned with the P1 helix and selectivity filter (rmsd = 4.10 Å). While the inserted direction of the transmembrane region of NMDAR is opposite to that of NavAb (Fig. 1b, c), the pore domains of these channels are well overlapped (Fig. 1d and Supplementary Fig. 1b). Leu176 of NavAb well overlapped with Asn616 of NR1 involved in the functional inhibition by magnesium ion of NMDAR[4,5,7] (Fig. 1d). The superimposition onto NMDAR NR2b subunit was lesser matched (rmsd = 5. 44 Å), and Leu176 of NavAb located between Asn615 and Asn616 of NR2b subunit (Fig. 1d). These residues were also involved in magnesium inhibition[4,5,7]. The AMPA receptor (AMPAR) is an ionotropic receptor similar to NMDAR. Superimposing of NavAb onto AMPAR, the RMSD is 5. 44 Å, which is lesser than that of NMDAR NR1. The AMPAR residue corresponding to Leu176 of NavAb is Gln586.

### Calcium block on Leu176 mutants

First, the wild-type channel of NavAb is activated even at very negative membrane potentials and requires a −240 mV holding potential for recovery[19,24]. A negative holding potential as deep as −240 mV makes it hard to evaluate the channel properties. Previous studies have shown that the N49K mutation in the voltage sensor domain raises the activation potential, and the −140 mV holding potential can sufficiently maintain the channel function[24,25]. The mutational site of N49K is far from SF (Fig. 1b). Therefore, to provide a stable current, the N49K mutation of NavAb was introduced into all constructs of NavAb in this study.

The current of NavAb N49K mutant was not attenuated even if the calcium concentration increased from 2.5 mM to 40 mM in all

depolarizing membrane potential (Fig. 1e and Supplementary Fig. 2a). To reproduce the divalent cation-blocking effect on NavAb N49K, we introduced L176N (L176N[NK]) and L176Q (L176Q[NK]) mutations which imitate the selectivity filter of NMDAR and AMPAR, respectively. They marked a significant decrease in sodium current according to extracellular calcium concentration. (Fig. 1f, g). Unlike the case of NMDAR[2], these calcium-blocking effects were observed in all depolarizing membrane potentials (Supplementary Fig. 2d, g). The positive shift of the reversal potential was slight when extracellular calcium ion concentration increased (Supplementary Fig. 2b, e, h). It indicates the low calcium permeability of L176N[NK] and L176Q[NK] mutants and the calcium ion selectivity against sodium ion ($P_{Ca}/P_{Na}$) of BacNavs[14,19]. Under higher extracellular calcium ion concentrations, the voltage dependency of activation was slightly shifted positively (Supplementary Fig. 2c, f, i), as in the case of NavPp[18]. The properties of these NavAb are different from those of NavPp.

We can find that the residue corresponding to Leu 176 has a wide variation in the selectivity filter of the tetrameric cationic channel[26]. The corresponding residue is phenylalanine in some BacNav and AnclNav (Supplementary Fig. 1a). L176F[NK] mutant showed no current reduction (Fig. 2a, and Supplementary Fig. 3a–c). It indicates that the L176F mutation does not generate the divalent cation-blocking property. EukCat, single-celled eukaryotic algae, has more variations of the corresponding residue of Leu176[26,27] (Supplementary Fig. 1a). Referring to the selectivity filter sequence of EukCat, we introduced several single mutations into the 176th residue and evaluated the effect on divalent cation block (Fig. 2). The L176A[NK] mutant showed the strongest calcium block (Fig. 2b and Supplementary Fig. 4a–c). The current of the L176A[NK] mutant almost disappeared in the 40 mM extracellular calcium condition. The L176G[NK] mutant showed only a half reduction

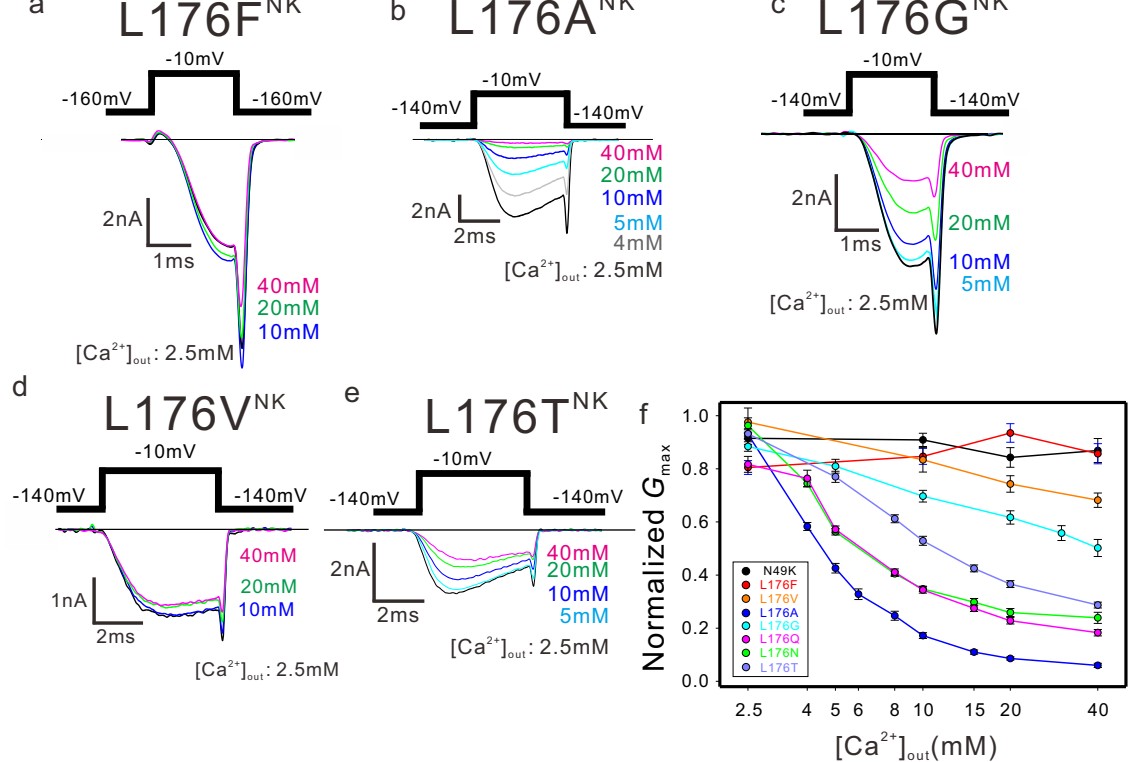

**Fig. 2 | The effect of Leu176 mutations on Ca²⁺ blocking. a–e** Representative current traces of NavAb L176F[NK], L176A[NK], L176G[NK], L176V[NK], and L176T[NK] mutants were generated by −10 mV stimulation pulses in various extracellular Ca²⁺ concentrations, respectively. **f** $G_{max}$ of the sigmoidal fitting curve of each outside calcium ion concentration condition of NavAb Leu176[NK] mutants normalized by the tail current generated by −10 mV stimulation pulse under

2.5 mM extracellular Ca²⁺ condition. Symbols and error bars indicate the value and the standard deviation of $G_{max}$ of the activation fitting curves in Supplementary Figs. 2–5, respectively. The data were obtained from biologically independent cells (n = 4; N49K, 5; L176N[NK], 5; L176Q[NK], 6; L176F[NK], 4; L176A[NK], 4; L176G[NK], 3; L176V[NK], and 4; L176T[NK]). The sample numbers of mutants are also listed in Supplementary Figs. 2–5.

of current in 40 mM extracellular calcium condition (Fig. 2c and Supplementary Fig. 4d–f). L176V[NK] showed little current reduction (Fig. 2d, and Supplementary Fig. 5a–c). Therefore, smaller side chain mutants can generate the divalent cation-blocking property. L176T[NK] mutant, a hydrophilic mutant like L176Q[NK] and L176N[NK] mutants, also showed calcium block (Fig. 2e and Supplementary Fig. 5d–f). The blocking extent of calcium ions on the L176T[NK] mutant is similar to those of L176Q[NK] and L176N[NK] mutants (Fig. 2f). EukCat has not been investigated for bivalent cation block. However, some EukCat may show the divalent cation-blocking property considering the amino-acid sequence. Since BacNav and EukCat are considered primitive channels, these results suggested that these mutations mimic the process of acquiring the divalent cation block that would have occurred in the early stages of channel evolution.

The calcium-blocking mutants also showed the current blocking in each magnesium and strontium condition (Supplementary Fig. 6). Therefore, this blocking effect would be common to divalent cations. These results suggested that the small or hydrophilic mutations at the 176th residue introduce the divalent cation block into NavAb[NK]. Therefore, we hypothesized that these mutations cause new interactions with divalent cations at the bottom of the selective filter. To elucidate this mechanism, we proceeded with structural analyses of these mutants.

## The structures of L176Q[NK] and L176G[NK] mutants

To investigate the structural changes of the calcium-blocking mutants, we crystallized and determined the crystal structures of L176Q[NK], L176G[NK], and N49K mutants with and without calcium ions at resolutions of approximately 3 Å. The asymmetric units and crystal lattice of these crystals were almost the same as those in our previous study (Supplementary Table 1)[24]. The electron density fitted well to the mutated side chain of the 176th residue of L176Q[NK] and L176G[NK] mutants (Fig. 3a–c and Supplementary Fig. 7 and Supplementary Movies 1–6). Because the solubilized proteins of L176N[NK], L176T[NK], and L176A[NK] mutants were unstable, we could not obtain their well-diffracted crystals. NavAb has three ion interaction sites in the selectivity filter, a high-energy-field site (Site_HFS), a centre site (Site_CEN), and an inner site (Site_IN) (Fig. 3d)[19]. For all three mutants, the electron density around all three sites increased in the calcium condition (Supplementary Fig. 7 and Supplementary Movies 1–6).

For evaluating the electron density, we compared difference maps between structures of each mutant with and without calcium ions

(Figs. 3d, 4b, d: green mesh). In the ionic pathway of the N49K mutant, there is no differential density of the calcium ion condition subtracted with the electron density of the no calcium ion condition (Fig. 3d). In the L176Q[NK] mutant, significant increases of differential density are observed between Site_CEN and Site_IN, and below Site_IN (Fig. 4b). These increased densities are due to calcium ions which cause the current inhibition. The amine group of the mutated L176Q side chain forms two weak hydrogen bonds with the main chain of Val173 and Met174 of the P1 helix (Fig. 4a, b). These interactions make the carbonyl group of the L176Q side chain face the centre of the ion pathway. By comparing the electron density of the L176Q[NK] and N49K mutants for the same ionic condition, we can find that the L176Q side chain is responsible for the increase in electron density below Site_IN with and without calcium ion (Fig. 4a, b: brown mesh). In the non-calcium condition, the differential density at Site_IN of the L176Q[NK] mutant seems to interact with two oxygen atoms of the carbonyl group of the L176Q side chain and Thr175 main chain (Fig. 4a). It suggested that the L176Q[NK] mutant can stabilize sodium or water molecule in this site more than N49K mutant. In the calcium condition, the differential density of Site_CEN is more significant than that of the NavAb N49K mutant as well as that of Site_IN (Fig. 4b). The stabilization effect is stronger for more positively charged calcium ions than that of sodium ions. Therefore, calcium ions stacked at the centre of the ion permeation pathway and blocked the current in the L176Q[NK] mutant.

In the case of the L176G[NK] mutant, the differential density increases because the glycine mutation creates an extra cavity at the interface between the selective filter and the inner vestibule (Fig. 4c, d). In non-calcium ion condition, the observed additional electron density of the extra cavity was at 3.7 Å from Cα carbon of L176G residue and assigned as a water molecule (Fig. 4c). The additional water molecule is close to five oxygen atoms of the main chain of Val173, Met174, Thr175, L176G and adjacent subunit Thr175 (Fig. 4c) in the non-calcium condition. The electron density at Site_CEN and Site_IN slightly increased in the calcium condition than in the non-calcium ion condition (Fig. 4d: green mesh). The electron density of Site_IN of the L176G[NK] mutant was also more substantial than that of the NavAb N49K mutant in calcium ion condition (Fig. 4d). This additional electron density is connected to the electron density assigned as the additional water molecule (Fig. 4d). Therefore, it is suggested that calcium ions stacked around Site_IN and blocked the current in the L176G[NK] mutant.

The electron densities of calcium-blocking mutants suggested that the calcium ions corresponding to the electron density beneath

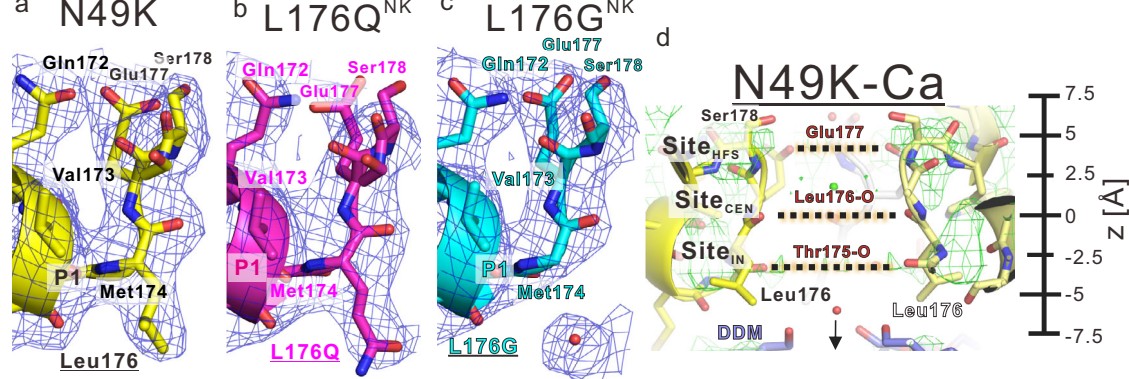

**Fig. 3 | The crystal structure of NavAb N49K, L176Q[NK], and L176G[NK] mutant.**
**a–c** The electron density around the 176th residue of the selectivity filter in the calcium condition. The cartoon view and carbon atoms of NavAb N49K, L176Q[NK], and L176G[NK] mutant are coloured yellow, magenta, and cyan, respectively. The $2F_O - F_C$ electron density map contoured at 1σ (blue mesh) shows the difference in the electron density of the mutated 176th residue. **d** Horizontal view of the differential electron density map of NavAb N49K mutant in the calcium conditions subtracted with that in the non-calcium. The upside is the extracellular side. Black dashed lines between Site_HFS, Site_CEN, and Site_IN indicate the interaction site of the selectivity filter, high-field-strength site, centre site, and inner site, respectively. Green mesh indicates the differential electron density map contoured at 5σ of N49K mutant $F_O$ in calcium condition subtracted with that in non-calcium condition.

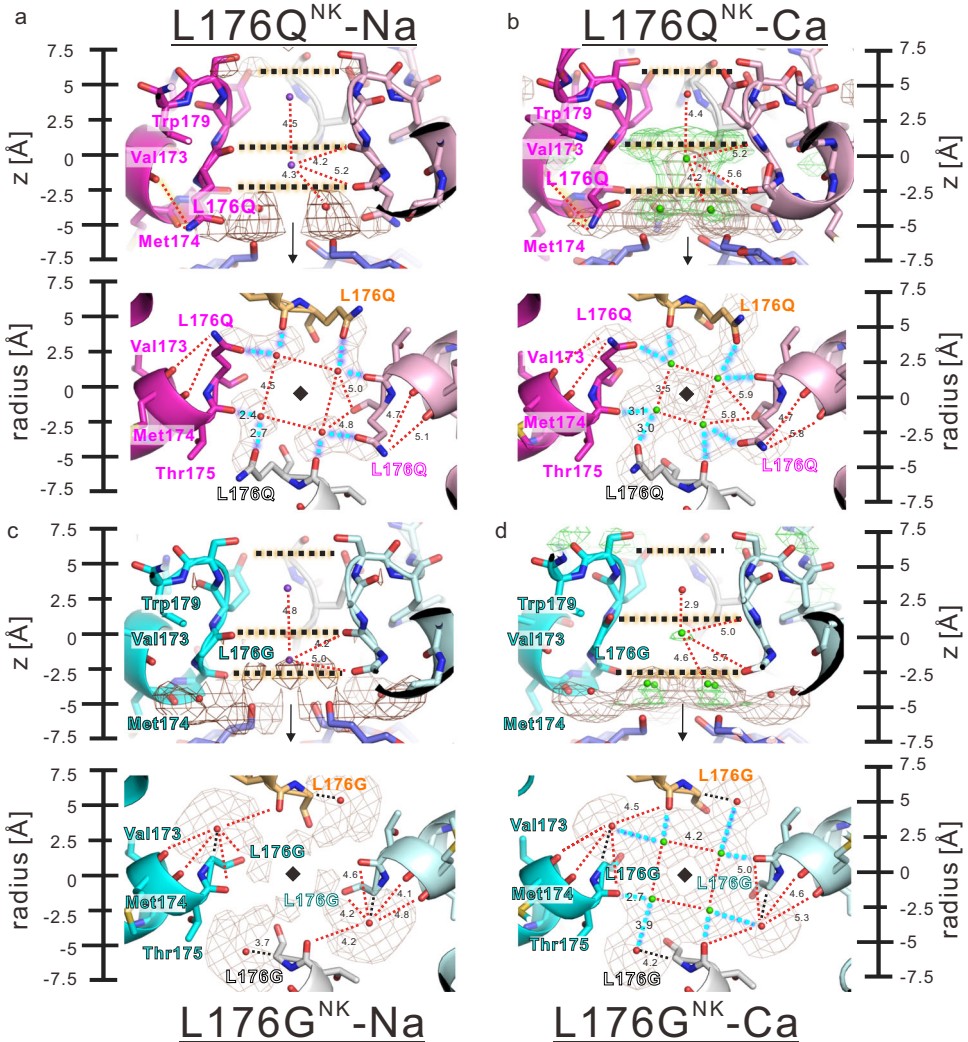

**Fig. 4 | The selectivity filter structure of NavAb L176Q$^{NK}$, and L176G$^{NK}$ mutant.** **a**, **b** Horizontal and vertical view of the ion pathway of NavAb L176Q$^{NK}$ mutant with the differential electron density map in the non-calcium and calcium conditions, respectively. Upper panels: The upside is the extracellular side. Lower panels: Vertical view of the ion pathway. Green mesh indicates the difference-electron density map contoured at 5σ of each mutant $F_O$ in calcium condition subtracted with that in non-calcium condition. Brown mesh indicates the differential electron density map contoured at 5σ of each mutant $F_O$ subtracted with N49K mutant $F_O$ in the same ionic condition. Cyan dashed lines indicate the interaction between the L176Q side chain and calcium ions at the centre of the ion pathways. Red dashed lines indicate the hydrogen bond between the mutated residue and P1 helix residues. The cartoon view and carbon atom of the four NavAb L176Q$^{NK}$ subunits are coloured magenta, white, pale magenta, and orange, respectively. The diamond and arrow indicate the four-hold rotation axis. The unit of numerical value described in the interaction is Angstrom. **c**, **d** Horizontal and vertical view of the ion pathway of NavAb L176G$^{NK}$ mutant with the differential electron density map in the non-calcium and calcium conditions, respectively. Upper panels: The upside is the extracellular side. Lower panels: Vertical view of the ion pathway. Similar to the case of the L176Q$^{NK}$ mutant, the interaction and electron density were noted. Additionally, black dashed lines indicate the distance from the Cα-carbon of L176G to the additional water molecule. The cartoon view and carbon atom of the four NavAb L176G$^{NK}$ subunits are coloured cyan, white, pale magenta, and orange, respectively.

the selectivity filter suppress the sodium current. The calcium ions in the selectivity filter have enough occupancy, but their temperature factors are higher than that of the surrounding protein atoms (Table 1). This may be because the calcium ions assigned in the selectivity filter were heavier than the actual atoms or that these calcium ions could move freely in the selectivity filter. These possibilities suggest that water and calcium ions are mixed in the selectivity filter, and that water and calcium ions can move freely around Site$_{IN}$. Due to the resolution limit of the crystal structure, it has been difficult to determine the number of calcium ions stacked in the selective filter. However, as many as four divalent cations are unlikely to stack at the bottom of the selectivity filter because of the positive charge repulsion. Therefore, it is appropriate for the observed electron density to be the averaged electron density of one or two calcium ions in the inner vestibule. Magnesium and strontium ions also blocked the calcium-blocking

mutants (Supplementary Fig. 6). The divalent cations might form loose and adjustable interaction networks with amino acids at the inner vestibule of the blocking mutants.

## Molecular dynamics simulation of the calcium-blocking mutants

We proceeded with MD simulation to analyze the detailed interaction between calcium ions and the mutated side chain of L176Q$^{NK}$ and L176G$^{NK}$ mutants. From these simulations, we can also gain insight into the behaviour of L176N$^{NK}$ and L176A$^{NK}$ mutants, whose structures were undetermined. Furthermore, we can deduce the number of calcium ions in the channel lumen and the generality of the calcium-blocking mechanism by comparing the results of these simulations.

We performed a non-equilibrium MD simulation by applying a −300-mV electric field to examine how ions permeate (Fig. 5a and

**Table 1 | The occupancy and B-factor of the molecules in the selectivity filter**

| z(Å) | | NavAb N49K | | L176Q$^{NK}$ | | L176G$^{NK}$ | |
|---|---|---|---|---|---|---|---|
| | | Na | Ca | Na | Ca | Na | Ca |
| 7.5 | Molecule (occupancy/B-factor) | | $H_2O$ (0.25/61.97) | | | | |
| 5 | Site$_{HFS}$ | | | The carbon of the 177th carboxyl group | | | |
| | | (1.00/53.03) | (1.00/42.17) | (1.00/16.17) | (1.00/19.77) | (1.00/94.29) | (1.00/36.17) |
| 2.5 | Molecule (occupancy/B-factor) | $Na^+$ (0.15/62.21) | $Ca^{2+}$ (0.2/85.85) | $Na^+$ (0.16/42.76) | $H_2O$ (0.25/25.10) | $Na^+$ (0.25/114.3) | $H_2O$ (0.25/97.41) |
| 0 | Site$_{CEN}$ | | | The oxygen of the 176th main chain carbonyl group | | | |
| | | (1.00/58.83) | (1.00/30.65) | (1.00/33.16) | (1.00/19.44) | (1.00/133.9) | (1.00/54.17) |
| −1.5 | Molecule (occupancy/B-factor) | | | $Na^+$ (0.25/67.47) | $Ca^{2+}$ (0.25/44.40) | $Na^+$ (0.25/149.1) | $Ca^{2+}$ (0.25/103.0) |
| −2.5 | Site$_{IN}$ | | | The oxygen of the 175th main chain carbonyl group | | | |
| | | (1.00/67.04) | (1.00/44.15) | (1.00/31.11) | (1.00/29.31) | (1.00/127.8) | (1.00/58.30) |
| −5 | Molecule (occupancy/B-factor) | | | The oxygen of the L176Q side chain carbonyl group | | $H_2O$ (1.00/114.6) | $H_2O$ (1.00/50.08) |
| | | | | (1.00/44.94) | (1.00/41.00) | | |
| | | | | $H_2O$ (1.00/41.54) | $Ca^{2+}$ (1.00/41.33) | | $Ca^{2+}$ (1.00/113.9) |

Supplementary Movies 7–18). We plotted the positions of ions along the z-axis (the axis of the channel pathway) against time as the ionic trajectory (Fig. 5b, c, and Supplementary Figs. 8, 9). In the first trial, calcium ions bound to the extracellular and cytosolic entrance of the channel and inhibited the sodium ion permeation even in the wild-type channel (Supplementary Fig. 8 and Supplementary Movies 7, 8). In solution, monovalent metal ions cannot exhibit all of their valences due to hydration since there is the so-called charge transfer[28]. This tendency is more pronounced with polyvalent ions. Considering this, for suitable parametrization of the Coulombic interaction with metal cations, we applied the electronic continuum correction (ECC)[29,30]. Applying the ECC, sodium ions were kept permeating, and calcium ions infrequently permeated the pore domain of the wild-type channel (Fig. 5b, c, and Supplementary Movies 9, 10). The ionic currents, estimated by the number of permeated ions over simulation time, of the wild-type channel slightly increased with calcium ions (Fig. 6a). The calculated current of wild-type NavAb is approximately 20 pA with symmetrical 100 mM NaCl under −300 mV membrane potential in this simulation (Fig. 6a and Table 2). These current values are consistent with the previously reported single channel recording experimental data of NavAb or BacNavs[31,32]. Therefore, the simulation applying the ECC well represented the actual situation.

Similar continuous permeations of sodium ions were observed in the calcium-blocking mutants without calcium ions (Supplementary Fig. 9 and Supplementary Movies 11, 13, 15, 17). With calcium ions (100 mM sodium ion and 40 mM calcium ion), the sodium ion permeation is significantly decreased in the calcium-blocking mutants (Figs. 5c, 6a, Table 2, and Supplementary Movies 12, 14, 16, 18) in the MD simulation. We frequently observed a calcium ion stacking at the bottom of the selectivity filter (z = −5 Å to 0 Å) in the calcium-blocking mutants (Fig. 5c). More than one calcium ion is never stacked in the inner vestibule (z = −15 Å to 0 Å) simultaneously. The increase in electron density around Site$_{CEN}$ and Site$_{IN}$ indicated by the crystal structure in the calcium ion condition (Fig. 4b, d) was thought to be due to a single calcium ion. It is considered the averaged electron density of one calcium ion and hydrated water molecules. When calcium ion was stacked at the bottom of the selectivity filter (Fig. 5c: right), sodium ions did not permeate through the pore domain (Fig. 5c: left).

**Analysis of the free energy of calcium and sodium ions in the inner vestibule**
We evaluated the behaviour of calcium and sodium ions in the inner vestibule with calculated free energies (Supplementary Fig. 10). The two-dimensional free energy landscapes ($G(r, z)$) were computed by the following equation: $G(r, z) = -k_B T \ln P(r, z)$, where $k_B$, $T$, $r$, and $P(r, z)$ are the Boltzmann constant, temperature, the perpendicular axis to z, and the probability of finding ions at $r$ and $z$, respectively. The $G(r, z)$ of calcium ions indicates that calcium ions stably locate on the intracellular side of the intracellular gate (Fig. 6b: z = −20 Å) in the case of all channels (Supplementary Fig. 10). Still, these calcium ions did not disturb the sodium ion permeation, because the current of wild type channel was not decreased in the calcium ion condition. Therefore, we focused on the behaviour of ions in the inner vestibule.

We calculated a one-dimensional free energy profile to reveal stable points or binding sites for ions in the inner vestibule during permeation. The one-dimensional free energy profiles were calculated by integrating a two-dimensional free energy landscape with respect to $r$ up to 10 Å from the centre of the channel pore (Fig. 6c–g, and Supplementary Figs. 11, 12). The standard deviation of the one-dimensional free energy profiles of cations in the wild-type channel suggested that the free energy of the calcium-free sodium ion showed sufficient convergence in the inner vestibule (Fig. 6b: z = 0 Å to −10 Å) and the bulk solution (Fig. 6b: z < −20 Å, z > 20 Å) (Supplementary Fig. 11c). On the other hand, the standard deviation for calcium ions was more widely spread in the bulk region than for sodium ions because of the small number of calcium ions in the calculation system (Table 2 and Supplementary Fig. 11d). Because the permeation of calcium ions rarely happened, the standard deviation of calcium ions' free energy was more significant in the inner vestibule than that of sodium ion in the calcium-free condition (Supplementary Fig. 11c, d). In the calcium ion condition, the standard deviation of sodium ions' free energy is also significant in the inner vestibule (Supplementary Fig. 11e). It indicates that the presence of calcium ions affects the behaviour of the sodium ions in the inner vestibule.

From the one-dimensional free energy profile, it isn't easy to discuss the sodium ion permeation mechanism from the extracellular entrance to the inner vestibule (z = 15 Å to −10 Å), where multiple sodium ions simultaneously can enter (Fig. 5b, c, Supplementary Fig. 9 and Supplementary Movies 9–18). However, in the intracellular gate (z = −10 Å to −20 Å), only one sodium ion can be placed at a time (Fig. 5b, c and Supplementary Fig. 9). The simulated sodium current of the divalent-cation-blocking mutants decrease under the calcium condition (Fig. 6a), which correlated with the increment of the one-dimensional free energy profiles of sodium ion around the intracellular gate under the calcium condition (Fig. 5c–g).

**The calcium-ion behaviour in the inner vestibule of the mutants**
In the L176Q and L176N mutants, calcium ions' free energy significantly decreases near the 176th carbonyl oxygen atom (z = 0 Å). This free

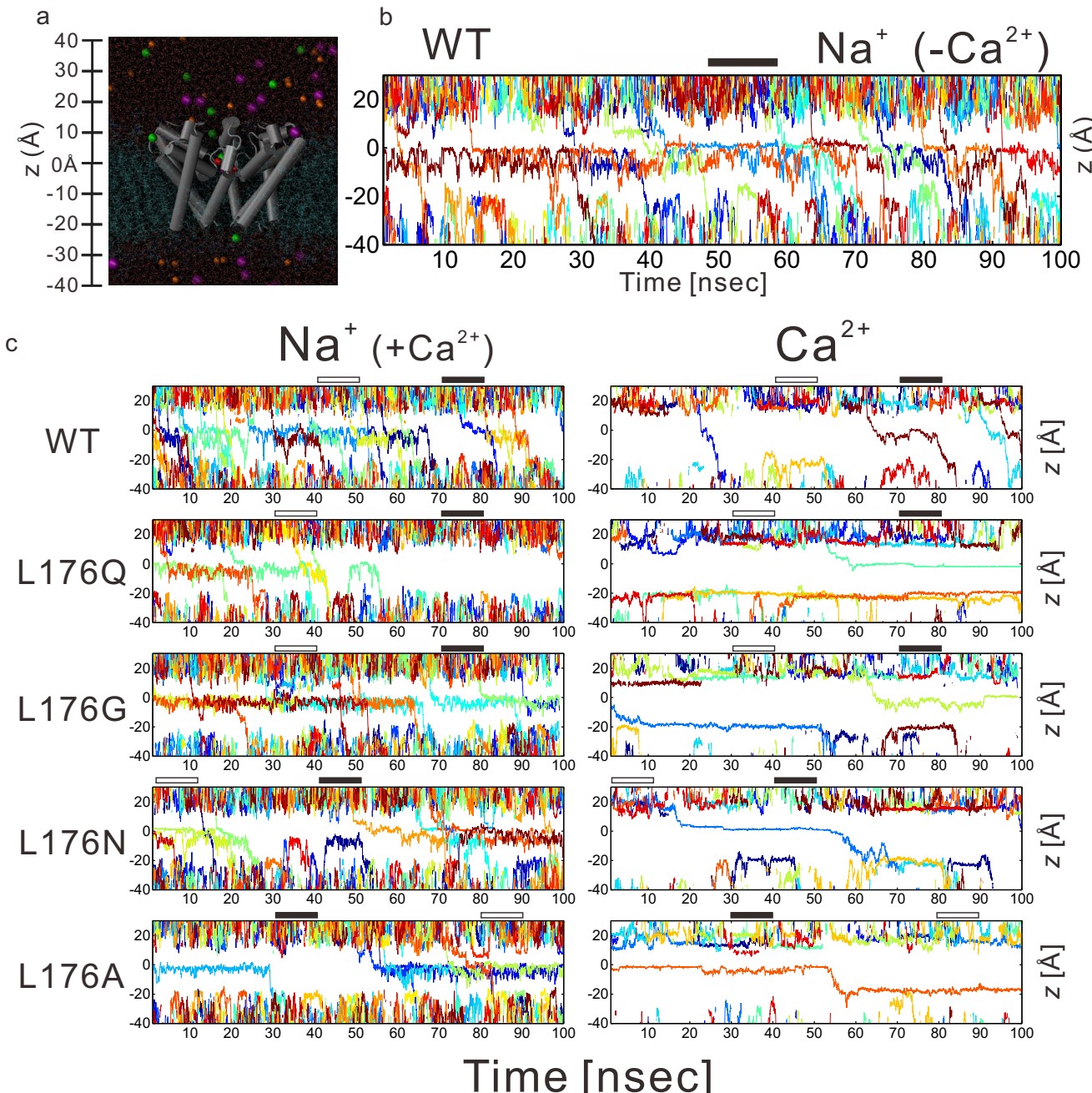

**Fig. 5 | The NavAb channel pore domain structure and the ion trajectories calculated in MD simulations. a** Structure of the pore domain of NavAb (grey cylinder) in a membrane lipid bilayer (cyan line). Residues Met130−Val213 in three subunits are shown. Sodium, chloride, and calcium ions are represented by purple, orange, and green spheres, respectively. Red spheres indicate the main chain carbonyl oxygen atoms of the 176th residue. The baseline of the z-axis is the average value of four z-axis values of the main chain carbonyl oxygen atoms of the 176th residue. **b** The sodium ion trajectories of a wild-type NavAb channel in calcium-free conditions. The trajectories of repeated ion permeation across the

pore are shown along the z-axis as a function of time (nsec). The black bars indicate the time period used for evaluating water density in Supplementary Fig. 13. Z-axis values correspond to that in Fig. 4a. **c** The sodium and calcium ion trajectories in calcium condition in each mutant channel. MD simulations generate the trajectories of each ion. The trajectories of repeated ion permeation across the pore are shown along the z-axis as a function of time (nsec). The black and white bars indicate the time period used for evaluating water density with or without stacking calcium ions in Supplementary Fig. 13, respectively. Z-axis values correspond to that in Fig. 5a.

energy minimum is consistent with the increment of the electron density observed in the L176Q mutant crystal structure under the calcium condition (Fig. 4b). Following the calcium ion's presence, sodium ions' free energy is increased from the intracellular ($z = −15$ Å) to the whole of the inner vestibule ($z = 0$ Å to $−10$ Å) of the L176Q mutant (Fig. 6d). It is evidence that calcium ions disturb sodium ions from entering the channel. In the case of the L176N mutant, the free

energy of sodium ions is also increased at the bottom of the selectivity filter (Fig. 6f).

As with calcium-free energies of L176Q and L176N mutants, those of L176G and L176A mutants decreased in the intracellular region. But the position of the minimums free energy was different ($z = −5$ Å) (Fig. 6d−h), which corresponds to the extra cavity created by the small side chain of 176th residue in the crystal structure (Fig. 4c, d). MD

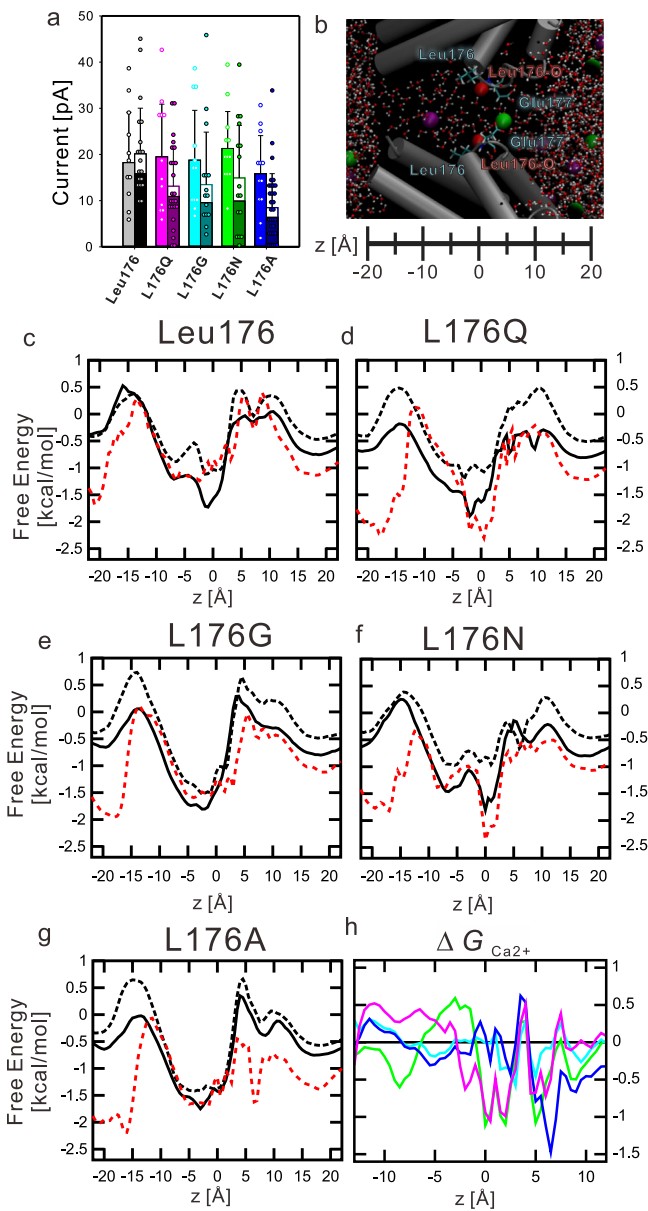

**Fig. 6 | The ionic current calculated from molecular dynamics simulation and a one-dimensional free energy landscape. a** Ionic current and standard deviation generated by the permeating ions in molecular dynamics simulations of each mutant. The left bars indicate the current generated by sodium ions in the non-calcium condition simulation (100 mM sodium ion). The right bars indicated the current generated by sodium and calcium ions in the calcium ion condition simulation (100 mM sodium ion and 40 mM calcium ion). Open parts and closed parts indicated calcium current and sodium current, respectively. Bar height and error bars indicate the average and the standard deviation of the generated currents calculated from the number of permeating ions in the 100 nsec molecular dynamics simulations, respectively. The open and closed dots indicate the ionic currents produced by the individual 100 nsec production runs. The sample numbers are the individual 100 nsec production runs ($n = 12$; Leu176 without Ca$^{2+}$, $n = 10$; Leu176 with Ca$^{2+}$, 12; L176Q$^{NK}$ without Ca$^{2+}$, $n = 24$; L176Q$^{NK}$ with Ca$^{2+}$, 12; L176G$^{NK}$ without Ca$^{2+}$, $n = 16$; L176G$^{NK}$ with Ca$^{2+}$, 12; L176N$^{NK}$ without Ca$^{2+}$, $n = 16$; L176N$^{NK}$ with Ca$^{2+}$, 12; L176A$^{NK}$ without Ca$^{2+}$, $n = 32$; L176A$^{NK}$ with Ca$^{2+}$ in Table 2). **b** Selectivity filter and inner vestibule of NavAb. The transmembrane and pore helices of NavAb (grey cylinder) form the inner vestibule. The ball-and-stick model indicates water molecules. The purple sphere indicates a sodium ion. The green sphere indicates a calcium ion. Red spheres indicate the main chain carbonyl oxygen atoms of the 176th residue. **c–g** Sodium ions' Z-axis free energy landscape in the inner vestibule of the wild-type, L176Q, L176G, L176N, and L176A channel, respectively. Black and dashed black lines indicated the free energy of sodium ions in calcium-free conditions and calcium conditions, respectively. The dashed red line indicates the free energy of calcium ions. **h** Z-axis free energy difference of calcium ions in the inner vestibule of each channel relative to the wild-type channel. Magenta, cyan, green, and blue lines indicated the free energy difference of calcium ions of L176Q, L176G, L176N, and L176A mutants, respectively.

simulation indicates that water molecules are also more likely to be present in the extra cavity of L176G and L176A mutants ($z = 0$ Å) (Fig. 7a–c and Supplementary Fig. 13), which is also consistent with the electron density of the crystal structure of the L176G mutants (Fig. 4c, d). Thus, hydrated calcium ions can be stably present at the selectivity filter connections in the lumen of L176G and L176A. In the case of K channels, the transition of potassium ions from the cytosol to the inner vestibule triggers the ion transition from the selectivity filter to the extracellular solution[33]. The stagnant of calcium ions in the inner vestibule would make it more difficult for sodium ions to enter the inner vestibule.

### The molecular mechanism of divalent cation block

Electrophysiology experiments indicated that divalent cation block is caused by two types of mutations: hydrophilic and smaller-side-chain mutations. The property of these two types of mutations is apparently different, and it was surprising that the same inhibitory effect appeared despite such differences. The crystal structures of these mutants show an increment of the electron density derived from calcium ions at the bottom of the selective filter (Figs. 3, 4). In the small-side-chain mutant, electron densities of additional water molecules were found in the

space created by the smaller-side-chain mutation. The MD simulation indicated only one calcium ion is in the pathway at a time, and it is enough for the divalent cation block. These results suggested that the mutations of different properties also provide similar inhibitory mechanisms (Fig. 8). Hydrophilic side-chain mutations provide hydrogen bonds to molecules around the Site$_{IN}$ of the selective filter, allowing calcium ions to stack at Site$_{CEN}$, the centre of the ion permeation pathway (Fig. 8: middle). Small-side-chain mutation of Leu176 creates extra cavities where the original leucine side chains were. Including additional water molecules here allows divalent cations to interact with water molecules and block the entrance to the inner vestibule (Fig. 8: right), similar to hydrophilic mutations on the residues.

It is still challenging to reproduce magnesium blocking of NMDA receptors in NavAb mutants. The magnesium blocking the NMDA receptor is removed during the depolarization state[2]. In our case, the calcium blocking remains regardless of the potential. This mismatch should be an issue that must be solved in the future. BacNavs are the most advanced channel groups with well-organized structural analysis and simulation. Therefore, our results and methods of structural analysis and MD simulations are also expected to play an active and meaningful role in the advanced analysis of a divalent cation-blocking mechanism.

## Methods

### Site-Directed mutagenesis and construction of NavAb mutants

The NavAb mutated DNAs were subcloned into the modified pBiEX-1 vector (Novagen; 71234-3CN) that was modified by replacing the fragment from NcoI site (CCATGG) to SalI site (GTCGAC) in multicloning site with the sequence "CCATGGGCAGCAGCCATCATCATCATCATCA-CAGCAGCGGCCTGGTGCCGCGCGGCAGCCATATGCTCGAGCTGGTGC-CGCGCGGCAGCGGATCCTAAGTCGAC"[24]. To add the N-terminal His tag and thrombin cleavage site, the NavAb mutated DNAs were subcloned between BamHI and SalI site. The polymerase chain reaction accomplished site-directed mutagenesis using PrimeSTAR® Max DNA Polymerase (Takara bio; R045A). We used two primers for making mutants: CAAGTTATGACTnnbGAATCTTGGTCAATGGGTATTGTCAGAC as forward primer, and GACCAAGATTCvnnAGTCATAACTTGAAATAGTGT

**Table 2 | The condition of molecular dynamics simulation**

| | WT no ECC | WT | | L176Q | | L176G | | L176N | | L176A | |
|---|---|---|---|---|---|---|---|---|---|---|---|
| Sodium ion (mM) | 100 | 100 | 100 | 100 | 100 | 100 | 100 | 100 | 100 | 100 | 100 | 100 |
| Calcium ion (mM) | – | 40 | – | 40 | – | 40 | – | 40 | – | 40 | – | 40 |
| Calculation time (µsec) | 1.2 | 1.2 | 1.2 | 2.0 | 1.2 | 2.4 | 1.2 | 1.6 | 1.2 | 1.6 | 1.2 | 3.2 |
| Sodium ions | 25 | 25 | 25 | 25 | 25 | 25 | 25 | 25 | 25 | 25 | 25 | 25 |
| Calcium ions | – | 10 | – | 10 | – | 10 | – | 10 | – | 10 | – | 10 |
| Chloride ions | 13 | 33 | 9 | 29 | 9 | 29 | 9 | 29 | 9 | 29 | 9 | 29 |
| Water molecules | 13684 | 13684 | 13684 | 13684 | 13684 | 13684 | 13684 | 13684 | 13684 | 13684 | 13684 | 13684 |
| Number of lipids | 210 | 210 | 210 | 210 | 210 | 210 | 210 | 210 | 210 | 210 | 210 | 210 |
| No. of sodium ion permeation per 100 nsec | 10.29 | 1.54 | 11.38 | 10.53 | 12.17 | 6.81 | 11.71 | 5.23 | 13.29 | 6.19 | 9.88 | 3.98 |
| Standard deviation of permeation | 1.062 | 0.48 | 0.84 | 1.63 | 2.85 | 2.56 | 2.79 | 2.53 | 1.46 | 0.71 | 1.33 | 1.67 |
| No. of calcium ion permeation per 100 nsec | – | 0.08 | – | 1.28 | – | 0.69 | – | 1.20 | – | 1.56 | – | 0.66 |
| Standard deviation of permeation | – | 0.12 | – | 0.69 | – | 0.16 | – | 0.61 | – | 0.60 | – | 0.34 |

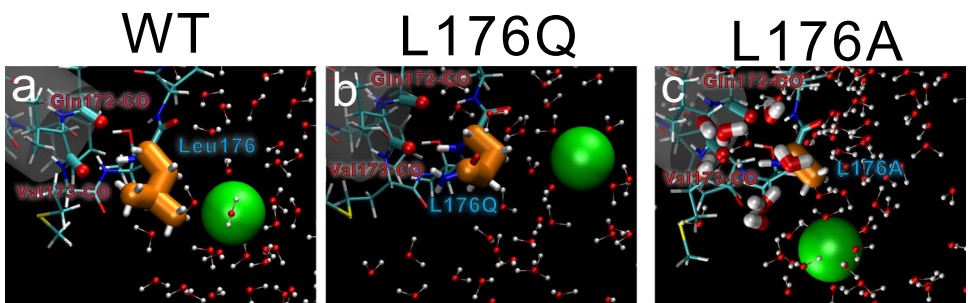

**Fig. 7 | Representative coordinates of MD calculation and the proposed molecular model of the divalent cation blocking. a–c** Representative coordinates of the P1-helix C-termini of NavAb wild-type, L176Q, and L176G, respectively. Green spheres indicate calcium ions. Cylinder and cyan sticks indicate the P1 helices and these residues. Orange fat sticks indicate the 176th residue. Ball-and-stick models indicate water molecules. Tube models indicate the water molecules within 5 Å of the main chain oxygen atoms of Gln172 and Val173.

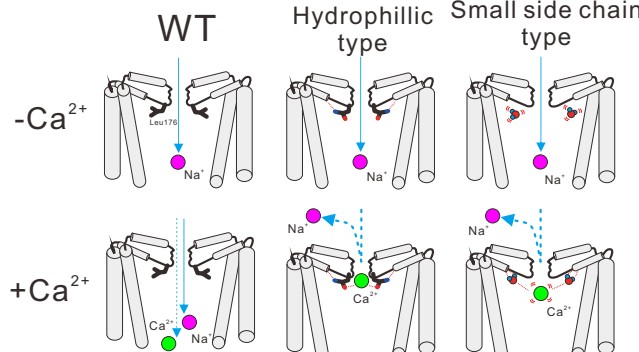

**Fig. 8 | The proposed molecular model of NavAb wild-type, L176Q, and L176G, respectively.** The front and rear subunits were removed for clarity. Grey cylinders indicate the helices of the NavAb pore domain helices. Green and purple spheres indicate calcium and sodium ions, respectively.

ATAAAAAG as revers primer. The primers were synthesized by Fasmac corp. All clones were confirmed by DNA sequencing.

## Protein expression and purification

Proteins were expressed in the *Escherichia coli* KRX strain (Promega; L3002). Cells were grown at 37 °C to an OD$_{600}$ of 0.6, induced with 0.2% α-L (+)-Rhamnose Monohydrate (Fujifilm Wako; 182-00751), and grown for 16 h at 25 °C. The cells were suspended in TBS buffer (20 mM Tris-HCl pH 8.0, 150 mM NaCl) and lysed using a French Press (SLM AMINCO) at 12,000 psi. Low-speed centrifugation removed cell debris (12,000 × g, 30 min, 4 °C). Membranes were collected by centrifugation (100,000 × g, 1 h, 4 °C) and solubilized by homogenization in TBS buffer containing 30 mM n-dodecyl-β-D-maltoside (DDM, Anatrace; D310). After centrifugation (40,000 × g, 30 min, 4 °C), the supernatant was loaded onto a HIS-Select® Cobalt Affinity Gel column (Sigma-Aldrich; H8162-100ML). The protein bound to the cobalt affinity column was washed with 10 mM imidazole in TBS buffer containing 0.05% lauryl maltose neopentyl glycol (LMNG, Anatrace; NG310) instead of DDM. After washing, the protein was eluted with 300 mM imidazole, and His tag was removed by thrombin digestion (overnight, 4 °C). Eluted protein was purified on a Superdex-200 column (Cytiva; 28990944) in TBS buffer containing 0.05% LMNG.

## Crystallization and structural determination

Before crystallization, the purified protein was concentrated to ~20 mg ml$^{-1}$ and reconstituted into a bicelle solution[34], containing a 10% bicelle mixture at 2.8:1 (1,2-dimyristoyl-sn-glycero-3-phosphorylcholine [DMPC, Anatrace; D514]: 3-[(3-cholamidopropyl) dimethylammonio]−2-hydroxypropanesulfonate [CHAPSO, DOJINDO; C020]). The NavAb-bicelle preparation was mixed in a 10:1 ratio. Prepared proteins were crystallized by sitting-drop vapour diffusion at 20 °C by mixing 300 nl volumes of the protein solution (8–18 mg/ml) and the reservoir solution (9–11% polyethylene glycol monomethyl ether [PEG MME] 2000 (Hampton; HR2-613), 100 mM sodium chloride, 100 mM magnesium nitrate, 25 mM cadmium nitrate and 100 mM Tris-HCl, pH 8.4) with mosquito LCP (TTP Labtech). The crystals were grown for 1 to 3 weeks, and after growing, the crystals were transferred into the reservoir solution without cadmium nitrate, which was replaced with the following cryoprotectant solutions. The cryoprotectant solution of the non-calcium condition contains 11% PEG MME 2000, 100 mM Tris-

HCl pH 8.4, 2.5 M sodium chloride, and 20% (v/v) DMSO. The cryo-protectant solution of the calcium condition additionally contains 100 mM calcium nitrate.

All data were collected at BL41XU, BL45XU, and BL32XU of SPring-8 and merged using the automatic data processing system KAMO[35] with XDS[36]. The data sets of NavAb N49K and L176G[NK] were obtained from a single crystal in both the calcium condition and the non-calcium condition. The data collections of NavAb L176Q[NK] were performed automatically using the ZOO system[37], and the data sets were obtained from four crystals in both calcium and non-calcium conditions. Analyses of the data with the University of California Los Angeles anisotropy server[38] revealed severely anisotropic crystals. Therefore, the data sets were ellipsoidally truncated and rescaled to minimize the inclusion of poor diffraction data.

A molecular replacement method with PHASER[39] provided the initial phase using the structure of NavAb N49K mutant (pdb code; 5yuc) as the initial model. The final model was constructed in COOT (version 0.9.2)[40] and refined in refmac5[41] and Phenix (version 1.18)[42]. CCP4 package (v.7.0.078)[43] was used for Structural analysis. Supplementary Table 1 summarizes Data collection and refinement statistics of all crystals. All figures in the present paper were prepared using the program PyMOL 2.5.2[44].

### Electrophysiological measurement in insect cells

The recordings were performed using SF-9 cells. SF-9 cells (ATCC catalogue number CRL-1711) were grown in Sf-900™ II medium (Gibco; 10902096) complemented with 0.5% 100× Antibiotic-Antimycotic (Gibco; 15240062) at 27 °C. Cells were transfected with target channel-cloned pBiEX vectors and enhanced green fluorescent protein (EGFP)-cloned pBiEX vectors using Fugene HD transfection reagent (Promega; E2311). First, the channel-cloned vector (1.5 µg) was mixed with 0.5 µg of the EGFP-cloned vector in 100 µL of the culture medium. Next, 3 µL Fugene HD reagent was added, and the mixture was incubated for 10 min before the transfection mixture was gently dropped onto cultured cells. After incubation for 16–48 h, the cells were used for electrophysiological measurements.

For measurement of the calcium blocking, pipette solution (140 mM CsF, 10 mM NaCl, 10 mM EGTA, and 10 mM HEPES [pH 7.4 adjusted by CsOH]) was used for the current measurement in the same way as the measurement of the calcium blocking of NavPp[18]. As a bath solution, calcium-blocking starting solution (30 mM NaCl, 120 mM NMDG-Cl, 2.5 mM CaCl$_2$, 10 mM HEPES [pH 7.4 adjusted by NaOH], and 10 mM glucose) was used for the 2.5 mM Ca$^{2+}$ blocking condition. In each calcium condition, 10 mM CaCl$_2$ was replaced per 15 mM NMDG-Cl. To measure the magnesium and strontium blocking, starting solution (30 mM NaCl, 120 mM NMDG-Cl, 2.5 mM CaCl$_2$, 10 mM HEPES [pH 7.4 adjusted by NaOH], and 10 mM glucose) was used for the 0 mM magnesium or strontium blocking condition. For the magnesium and strontium blocking, 10 mM MgCl$_2$ and SrCl$_2$ were replaced per 15 mM NMDG-Cl. Cancellation of the capacitance transients and leak subtraction were performed using a programmed P/10 protocol delivered at −140 mV. The bath solution was changed using the Dynaflow® Resolve system. All experiments were conducted at 25 ± 2 °C using a whole cell patch clamp recording mode with a HEKA EPC 10 amplifier and Patch master data acquisition software (v2x73). Data export was done using IGOR 6.37 and NeuroMatic (version 3.0b)[45]. All sample numbers represent the number of individual cells used for each measurement. Cells that had a smaller leak current than 0.5 nA were used for measurement. When any outliers were encountered, these outliers were excluded if any abnormalities were found in other measurement environments and were included if no abnormalities were found. All results are presented as mean ± standard error. The graph data were plotted using SigmaPlot 14.

### Molecular dynamics simulation

All simulations were performed using the MD program NAMD (version 2.9)[46]. The 5YUC structure was obtained from the RBSC PDB databank and used as the pore domain structure of NavAb, which was truncated to residues Met130-Val213. The side chains of Glu177 of all monomers were deprotonated. Using the NAMD program and membrane plugin version 1.1 of VMD (https://www.ks.uiuc.edu/Research/vmd/plugins/membrane/), 1-Palmitoyl-2-oleoyl-sn-glycero-3-phosphocholine (POPC) lipids were arranged to 98.0 × 98.0 Å$^2$ lipid bilayer, and water molecules were added into the lipid bilayer and formed as 98.0 × 98.0 × 75.0 Å$^3$ box system (APL = 70.1). The pore domain was embedded into the POPC membrane using membrane plugin version 1.1 of VMD. After embedding, the system was equibilized to 91.0 × 91.0 × 86.0 Å$^3$ box system by 100 ns calculation. Twenty-five sodium ions, 10 calcium ions, and 29 chloride ions were added into the system as the condition containing 100 mM NaCl and 40 mM CaCl$_2$ using Autoionize Plugin, Version 1.5 VMD (http://www.ks.uiuc.edu/Research/vmd/plugins/autoionize/). Therefore, the wild-type NavAb system containing 100 mM NaCl and 40 mM CaCl$_2$ comprised one channel, 210 POPC lipids, 25 sodium ions, 10 calcium ions, 29 chloride ions, and 13684 water molecules (see Table 2 for other concentrations). A periodic boundary condition of 91.0 × 91.0 × 86.0 Å$^3$ was imposed.

The force fields of CHARMM36[47] were used for the protein and lipid, and TIP3P[46] for the water. The sodium, calcium, and chloride parameters were applied with the electronic continuum correction[48,49]. Referring to the electronic continuum correction[29], we applied the electric charges of sodium, calcium, and chloride ions as 0.75, 1.5, and −0.75, respectively.

After 1000 steps minimization, MD simulations were carried out without any applied electric field for 10 ns to equilibrate the systems in each condition. A time step was 2 fs per step. The bonds, including H atoms, were constrained using the SHAKE algorithm[50]. The anisotropic NPT ensemble was employed to keep the temperature at 300 K and pressure at 1 atm with the Langevin thermostat (damping coefficient of 1/ps)[50] and the Nosé-Hoover Langevin piston barostat (piston period of 200 fs and piston decay time of 50 fs)[51]. Electrostatic interactions were calculated using the particle mesh Ewald method[52] with a 1.2 nm cut-off in real space. The cut-off of the Lennard-Jones interaction was 1.2 nm.

The equilibrated systems were used for MD simulations in the presence of −300 mV membrane potential via a constant applied electric field [−0.04 kcal/ (mol × Å × e)] applied across the Z-dimension of the simulation cell based on the equation: $V(mV) = Ez × Lz(Å) × 43.37$ where $Ez$ as the z-axis electric field, and $Lz$ as the z-axis length of the simulation cell. The numerical value of 43.37 was the force conversion coefficient used by NAMD. MD simulations were performed for four initial configurations at all conditions. The calculation time of all simulations was summarized in Table 2. The coordinates were visualized with VMD[53]. The graph data were plotted using SigmaPlot 14 and Gnuplot5.4.2.

### Reporting summary

Further information on research design is available in the Nature Portfolio Reporting Summary linked to this article.

## Data availability

The data that support this study are available from the corresponding authors upon request. The structural data generated in this study have been deposited in the Protein Data Bank under accession codes 8H9O (NavAb N49K mutant in sodium), 8H9W (NavAb N49K mutant in calcium), 8H9X (NavAb N49K/L176Q mutant in sodium), 8H9Y (NavAb N49K/L176Q mutant in calcium), 8HA1 (NavAb N49K/L176G mutant in sodium), 8HA2 (NavAb N49K/L176G mutant in calcium). The structure of NavAb N49K mutant for the initial model of molecular replacement

and the molecular dynamics simulation was available in the Protein Data Bank under accession codes 5YUC. The initial coordinates, input file, optimized parameter, and final coordinates of MD simulation are available via Figshare at [https://doi.org/10.6084/m9.figshare.23636175], reference number 23636175. The entire MD trajectory data are available from the corresponding author, K.I., upon request. The data that support the findings of this study are openly available in figshare at https://doi.org/10.6084/m9.figshare.23636175, reference number 23636175.

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

## Acknowledgements

The synchrotron radiation experiments were performed at BL41XU and BL32XU in SPring-8 with the approval of the Japan Synchrotron Radiation Research Institute (JASRI) (Proposal number 2016B2721, 2017B2735, and 2018B2710). We thank the beamline staff for their excellent facilities and support. We are appreciative of the English editing support program at the Clinical Study Support Center of Wakayama Medical University. The computation was performed using Research Center for Computational Science, Okazaki, Japan (Project: 22-IMS-C114, 22-IMS-C132 and, 23-IMS-C205). This research was partially supported by Platform Project for Supporting Drug Discovery and Life Science Research (Basis for Supporting Innovative Drug Discovery and Life Science Research [BINDS]) from AMED under Grant Number JP21am0101074 to A.O. This work was supported by Grants-in-Aid for Scientific Research (20H00451) and the Japan Agency for Medical Research and Development (AMED) to Y.F., 2022Wakayama Medical University Special Grant-in-Aid for Research Projects, Grants-in-Aid for Scientific Research (17K17795, and 20K09193), SEI Group CSR Foundation, Takeda Science Foundation, Institute for Fermentation to K. I., and World Premier International Research Center Initiative (WPI), MEXT, Japan to T.S.

## Author contributions

K.I. conceptualization; K.I., A.O., and Y.F. preparation and selection of experimental equipment; K.I and Y.O. performed protein purification; K.I. performed activity measurement, crystallization, crystal data collection, and crystallographic analysis; K.I. and Y.O. performed molecular dynamics simulation in early-stage study; K.I. and T.S. performed molecular dynamics simulation and data analysis; T.S. provided source codes for molecular dynamics simulation analysis; K.I., Y.O., and T.S. writing; K.I., T.S., A.O., and Y.F. funding acquisition; K.I. project administration.

## Competing interests

The authors declare no competing interests.
