## [Peer Review File · Nature Communications]

The structural basis of divalent cation block in a tetrameric prokaryotic sodium channelReviewers' Comments:

Reviewer #1:

Remarks to the Author:

The authors study the localization of divalent cation inhibition in a model mutated sodium channel using a combination of electrophysiology, x-ray crystallography and molecular dynamics simulations. It is found that, in contrast to the wildtype NavAb, mutations of Leu176 introduce calcium blocking. Both crystallography as well as MD simulations localize the calcium binding site as directly under the selectivity filter. The mutated residues either coordinate the calcium ion directly (Asn and Gln mutations) or create a cavity that allows binding of a hydrated calcium ion (e.g. Ala mutation). The results are interesting, the study well carried out and the paper is well written. I only have two minor concerns that should be addressed in a revised manuscript.

First, the statement "the currents with or without calcium ions are the same amount as the result of the electrophysiologic measurement (Fig.2f and Fig. 5c)." is confusing as neither Fig.2f nor Fig. 5c seem to show currents.

Second, the 1-dimensional free energy profiles are interesting, but it should be pointed out that, due to the multi-ion permeation mechanism, the 1-dimensional representation is only of limited use in the discussion of permeation (for Ca²⁺ binding it is probably OK).

Reviewer #2:

Remarks to the Author:

In the paper "The structural basis of the divalent cation blocking on tetrameric cation channel", Irie et al. address the question how divalent cations such as Ca²⁺ block ion conduction in cation channels. They use the bacterial Na⁺ channel NavAb as proxy for eukaryotic channels, introduce mutations in the selectivity filter region which lead to Ca²⁺ block, which is not observed in the WT NavAb channels. Using X-ray crystallography, they then study how WT and mutant channels differentially interact with Ca²⁺ ions. Finally, MD simulations are employed to gain additional insights. I am not an expert in MD simulations and will focus predominantly on the functional and structural analyses.

The methodology is appropriate for their study. The electrophysiological characterization of Ca²⁺ block is sufficient and convincing. X-ray crystallography also is the appropriate tool to address their question, as it allows to resolve electron density that stems from bound/coordinated ions much more reliably than cryo-EM. However, the resolution of all structures is between 3 and 3.5 Å which makes the interpretation of electron densities slightly harder. In general, the manuscript would benefit from more detailed explanations of the experiments, the reasoning behind, and the interpretations drawn from the results.

Overall, the study is conceptionally sound, however, considerable concerns arise with respect to novelty of the results, presentation of the structures, and interpretation of the data. As stated in the abstract, the proposed mechanism is not new. The authors claim that their data confirm previous hypothesis about Ca²⁺ block in Na⁺ channels. In my opinion they should not generalize their study to cation channels as done in the introduction.

- First, the authors propose that NavAb is a good model for NMDA and AMPA receptors, with respect to the pore, although the orientation in the membrane is different between these proteins (SF on the cytoplasmic or extracellular side, Fig 1a). However, the suggested similarity is not obvious from their structural comparison (Fig 1d,e). Colors need to be defined for both panels and, especially in Fig 1e., I cannot agree with NavAb being a suitable model. The structures, as presented, appear quite different (yellow vs magenta). The local RMSD in the SF regions need to be reported for NavAb-NMDAR and NavAb-AMAPR. While Ca²⁺ block seemingly is similar to the eukaryotic channels, in the conclusion the

authors state, that they are not able to capture all features. I recommend toning down the homology argument or to show more convincing structures and functional data.

- The authors start their results with NavAb N49K. No explanation is given for this mutation and the reader is left with confusion.

- Line 103-104: Reasoning for their mutations is only about side chain size and hydrophobicity/hydrophilicity. In the current state, conclusion about evolutionary processes in NavAb are unsubstantiated. At minimum, this would require sequence alignments based on evolutionary trees.

- Figure 3 presents x-ray structures of 3 NavAb proteins (WT, L176Q, and L176G) in different ionic conditions. Overall, the figure is hard to understand, especially panels d-i are overloaded with electron density in mesh representation. I suggest that the authors improve this figure by cleaning it up, which should help understanding their interpretations. For example, panels e, g, i could only show the difference maps. Especially in e, substantial densities are visible in the difference map what appears to be behind the filter, similar densities albeit less pronounced are found in g and i. This may point to other effects Ca²⁺ has on NavAb structure and function. More information and explanation of this observation is critically needed.

- Line 144 and 151: The authors describe interactions of the "carboxyl group" of the Q176 side chain. Glutamine does not have a carboxyl side chain! The electrostatics between Glu and Gln are different, since Gln (Q) is formally uncharged. Interaction with Ca²⁺ can only occur through lone ion pairs, not attractive forces between positive and negative charges.

- For all interactions described, distances need to be given in the figures. At this point it is not possible for the reader to judge the described interactions.

- Figure legend: line 618: "horizontal and vertical view of the ion pathway" – which one is it?

- In the last paragraph about the structure, the authors mention that it is unlikely for 4 Ca²⁺ ions to bind. The occupancies obtained from their x-ray structures need to be reported to make a case here. Again, in the current state the reader cannot evaluate the claims.

- I am not an expert in MD simulations. However, the data representation in Fig 4 and 5 should be improved and more detailed description should be added. Better explanation of the simulations and how energies were extracted will help the general reader to follow the line of argumentation. A structural representation along the z-axis (basically showing the ion conduction pathway) could make it easier to understand the details.

- Lines 279-281: "The blocking mechanism differs between mutations to hydrophilic residues and mutations to smaller residues (Fig.6e and f)." This is an important limitation that the authors point out themselves and contradicts the original assumption and the title. It directly shows that there is not a universal mechanism that explains Ca²⁺ block of Na⁺ channels. This also limits the approach and the model-nature of NavAb. The authors should be more careful in their generalization. This loops back to my earlier comment on homology between NavAb and NMDAR or AMPAR.

We are very appreciative of the reviewers' kind and thorough review. By revising according to reviewers' comments, we think we were able to clarify the points of discussion and eliminate unreasonable generalizations to make the content more widely accepted.

To avoid overgeneralization, the title has been changed to clarify that the inhibition mechanism in the newly introduced mutant has been analyzed.

"The reproduction and structural basis of the divalent cation blocking on tetrameric prokaryotic sodium channel."

Each movie corresponding to the trajectory in our manuscript was created and attached as supplementary movies.

To clarify Fig 3, we moved the electron density map to SupFig6 and depicted the difference Fourier map in Fig.3.

REVIEWER COMMENTS

Reviewer #1 (Remarks to the Author):

The authors study the localization of divalent cation inhibition in a model mutated sodium channel using a combination of electrophysiology, x-ray crystallography and molecular dynamics simulations. It is found that, in contrast to the wildtype NavAb, mutations of Leu176 introduce calcium blocking. Both crystallography as well as MD simulations localize the calcium binding site as directly under the selectivity filter. The mutated residues either coordinate the calcium ion directly (Asn and Gln mutations) or create a cavity that allows binding of a hydrated calcium ion (e.g. Ala mutation). The results are interesting, the study well carried out and the paper is well written. I only have two minor concerns that should be addressed in a revised manuscript.

First, the statement "the currents with or without calcium ions are the same amount as the result of the electrophysiologic measurement (Fig.2f and Fig. 5c)." is confusing as neither Fig.2f nor Fig. 5c seem to show currents.

I'm very sorry about our mistyping. "(Fig.2f and Fig. 5c)" was "(Fig.1f and Fig. 5a)" But, in response to the second point below, this description itself has been deleted.

Thanks to the points, we found that the legend of Figure 5a is small and easy to confuse. In Figure

5a, we described that the current value was converted from the number of permeating ions in the molecular dynamic simulation. So, the title of the Fig5 and the legend of Fig5a are shown as follows.

Figure 5. The ionic current is calculated from molecular dynamics simulation and a one-dimensional free energy landscape. a) Ionic current and standard deviation generated by the permeating ions in molecular dynamics simulations of each mutant. The left bars indicate the current generated by sodium ions in the non-calcium condition simulation (100 mM sodium ion). The right bars indicated the current generated by sodium and calcium ions in the calcium ion condition simulation (100 mM sodium ion and 40 mM calcium ion). Open parts and closed parts indicated calcium current and sodium current, respectively. Error bars indicate the standard deviations of the generated currents calculated from the number of permeating ions in the molecular dynamics simulations.

Second, the 1-dimensional free energy profiles are interesting, but it should be pointed out that, due to the multi-ion permeation mechanism, the 1-dimensional representation is only of limited use in the discussion of permeation (for Ca²⁺ binding it is probably OK).

Thank you very much for your critical comments. That's exactly what you pointed out, so we should pay close attention to the discussion of sodium permeation.

In response to your suggestions, we carefully rewrote the "**Analysis of the free energy of calcium and sodium ions in the inner vestibule.**"

We divided the part of this section that focuses on calcium free energy into the following new section.

The calcium-ion behavior in the inner vestibule of the mutants

Your suggestion clarified the content of the discussion and made it easier for general readers to understand. We are very appreciative again for your insightful comments.

Reviewer #2 (Remarks to the Author):

In the paper "The structural basis of the divalent cation blocking on tetrameric cation channel", Irie et al. address the question how divalent cations such as Ca²⁺ block ion conduction in cation channels. They use the bacterial Na⁺ channel NavAb as proxy for eukaryotic channels, introduce mutations in the selectivity filter region which lead to Ca²⁺ block, which is not observed in the WT NavAb channels. Using X-ray crystallography, they then study how WT and mutant channels differentially interact with Ca²⁺ ions. Finally, MD simulations are employed to gain additional insights. I am not an expert in MD simulations and will focus predominantly on the functional and structural analyses.

The methodology is appropriate for their study. The electrophysiological characterization of Ca²⁺ block is sufficient and convincing. X-ray crystallography also is the appropriate tool to address their question, as it allows to resolve electron density that stems from bound/coordinated ions much more reliably than cryo-EM. However, the resolution of all structures is between 3 and 3.5 Å which makes the interpretation of electron densities slightly harder. In general, the manuscript would benefit from more detailed explanations of the experiments, the reasoning behind, and the interpretations drawn from the results.

Overall, the study is conceptionally sound, however, considerable concerns arise with respect to novelty of the results, presentation of the structures, and interpretation of the data. As stated in the abstract, the proposed mechanism is not new. The authors claim that their data confirm previous hypothesis about Ca²⁺ block in Na⁺ channels. In my opinion they should not generalize their study to cation channels as done in the introduction.

Thank you for your assessment of the soundness of your research. Also, we are very appreciative of your precise comments. Finally, we are delighted that we can further increase the value of this paper by solving these problems. The responses to "the novelty of the results", "presentation of the structures", and "interpretation of the data" are described separately below.

"Interpretation of the data";

The issue with the interpretation of the data stems from overgeneralization. So, we changed the title to avoid generalizations. In addition, by adding "reproduction" and "tetrameric prokaryotic sodium channel", we clarified that the analysis is for the phenomenon generated in a specific channel. In the introduction, we have also revised the description to avoid generalizations. Significantly, we changed the overgeneralized last sentence of the Conclusion to one that mentioned that generalization to NMDAR still has challenges, as below.

It is still challenging to reproduce magnesium blocking of NMDA receptors in NavAb mutants. The magnesium blocking the NMDA receptor is removed during the depolarization state ². In our case, the calcium blocking remains regardless of the potential. This mismatch should be an issue that must be solved in the future. BacNavs are the most advanced channel groups with well-organized structural analysis and simulation. Therefore, our results and methods of structural analysis and MD simulations are also expected to play an active and meaningful role in the advanced analysis of a divalent cation-blocking mechanism.

"Presentation of the structures"

The presentation of the structures has certainly weaknesses and we are sorry for bothering you. Details will be described later, but we reconfigured and improved the difference map figures as Fig3 and the row-electron density map figures as SupFig7. In addition, movies on the crystal structure's raw electron density data and molecular dynamics simulation for ion permeation processes have been added as supplementary movies.

"The novelty of the results"

I'm very sorry about our poor description of the novelty of our study in our previous manuscript. The previous mutational analysis revealed the residues concerning the blocking. It, however, lacks a perspective on the molecular mechanisms of inhibition. In addition, crystal structure or cryo-electron microscopy did not directly observe the blocking divalent cations in the channel. The MD simulation only calculated the binding site of magnesium ions by structural minimization. There was no result of a dynamic inhibition mechanism based on an experimental structure.

Regarding these two cardinal unexplained points, such as not observing ions and not reproducing dynamic inhibition phenomena in simulation, our results provide clear answers, the electron densities in the Ca environment in the crystal structure, and the reproduction of the inhibition phenomenon in non-equilibrium MD simulation. In particular, there had been a significant problem reproducing divalent cations' behavior during MD simulation. The problem arises from the difficulty of handling the amount of charge of polyvalent metal ions. In this study, we solved this important problem. We think this point will greatly impact the MD simulation community.

I rewrote the abstract and introduction to clarify on these points.

- First, the authors propose that NavAb is a good model for NMDA and AMPA receptors, with respect to the pore, although the orientation in the membrane is different between these proteins (SF on the cytoplasmic or extracellular side, Fig 1a). However, the suggested similarity is not obvious from their structural comparison (Fig 1d,e). Colors need to be defined for both panels and, especially in Fig 1e., I cannot agree with NavAb being a suitable model. The structures, as presented, appear quite different (yellow vs magenta). The local RMSD in the SF regions need to be reported for NavAb-NMDAR and NavAb-AMAPR. While Ca²⁺ block seemingly is similar to the eukaryotic channels, in the Conclusion the authors state, that they are not able to capture all features. I recommend toning down the homology argument or to show more convincing structures and functional data.

Thank you for your precise point. The title has been changed so readers can understand that the molecular mechanism of inhibition by divalent cations reproduced in the tetrameric cation channel has been clarified.

"The reproduction and structural basis of the divalent cation blocking on tetrameric prokaryotic sodium channel."

In the wake of your suggestions, we reconsidered the superimposition. Until now, we had used only the P1 helix for superimposition, but we changed it to include P1-helix and a selectivity filter. We superimposed the diagonal subunits of NavAb onto NR1 (4.10Å), NR2B (5.44Å), and AMPA A2 (3.86Å) diagonal subunits. The number in parentheses reproduced RMSD. The RMSD of the selectivity filter has been added to the text. This improves the superimposition, so Fig1 has been updated. The superposition picture is easier to see using C α carbon traces for the half part and cartoon helix for the other half. We also added an alignment diagram of the area of superposition as SupFig1, by which readers might find the structural similarity. The entire pore domain of the superimposed is also shown in this figure. The discussion of homology is toned down throughout the manuscript. Color information is written in the figure.

Pore domain inner helix to outer helix alignment, including NavAb, BacNav, NMDAR, and AMPAR, added several EukCats to SupFig1. EukCats are a tetrameric channel group added for consideration of the evolutionary process, which will be described later,

- The authors start their results with NavAb N49K. No explanation is given for this mutation and the reader is left with confusion.

We immensely apologize for the less explanation about the N49K mutation. In the functional analysis of NavAb, it has become customary to include this mutation, and there has been little explanation. Therefore, regarding that reading by a wide audience, we added the following explanation of N49K in L98-L105.

First, the wild-type channel of NavAb is activated even at very negative membrane potentials and requires a -240mV holding potential for recovery^{19,24}. A negative holding potential as deep as -240mV makes it hard to evaluate the channel properties. Previous studies have shown that the N49K mutation in the voltage sensor domain raises the activation potential, and the -140mV holding potential can sufficiently maintain the channel function^{24,25}. The mutational site of N49K is far from SF (Fig. 1b). Therefore, to provide a stable current, the N49K mutation of NavAb was introduced into all constructs of NavAb in this study.

- Line 103-104: Reasoning for their mutations is only about side chain size and hydrophobicity/hydrophilicity. In the current state, Conclusion about evolutionary processes in NavAb are unsubstantiated. At minimum, this would require sequence alignments based on evolutionary trees.

We are very sorry that we did not provide enough information about the evolutionary process of tetrameric channels. Following the suggestion about the evolutionary phylogenetic tree, we quoted our recently written paper about the BacNav and various homotetrameric channels focused on the selectivity sequence. The eukaryotic homotetrameric channel, called EukCat, is important for considering the evolutionary process and is added to the text. We proceeded with the mutational analysis with the flow of variations of these sequences. As a result, the section on electrophysiology experiments of mutants was rewritten as L120-141 below.

We can find that the residue corresponding to Leu 176 has a wide variation in the selectivity filter of the tetrameric cationic channel²⁷. The corresponding residue is phenylalanine in some BacNav and AncINav (SupFig.1a). L176^F^{NK} mutant showed no current reduction (Fig. 2a, and Sup. Fig. 3a-c). It indicates that the L176F mutation does not generate the divalent cation-blocking property. EukCat, single-celled eukaryotic algae, has more variations of the corresponding residue of Leu176^{27,28} (SupFig.1a). Referring to the selectivity filter sequence of EukCat, we introduced several single mutations into the 176th residue and evaluated the effect on divalent cation blocking (Fig. 2). The L176^A^{NK} mutant showed the strongest calcium blocking (Fig. 2b and Sup. Fig. 4a-c). The current of the L176^A^{NK} mutant almost disappeared in the 40 mM extracellular calcium condition. The L176^G^{NK} mutant showed only a half reduction of current in 40 mM extracellular calcium condition (Fig. 2c and Sup. Fig. 4d-f). L176^V^{NK} showed little current reduction (Fig. 2d, and Sup. Fig. 5a-c). Therefore, smaller side chain mutants can generate the divalent cation-blocking property. L176^T^{NK} mutant, a hydrophilic mutant like L176^Q^{NK} and L176^N^{NK} mutants, also showed calcium blocking (Fig. 2e and Sup. Fig. 5d-f). The blocking extent of calcium ions on the L176^T^{NK} mutant is similar to those of L176^Q^{NK} and L176^N^{NK} mutants (Fig. 2f). EukCat has not been investigated for bivalent cation blocking. However, some EukCat may show the divalent cation-blocking property considering the amino-acid sequence. Since BacNav and EukCat are considered primitive channels, these results suggested that these mutations mimic the process of acquiring the divalent cation blocking that would have occurred in the early stages of channel evolution.

- Figure 3 presents x-ray structures of 3 NavAb proteins (WT, L176Q, and L176G) in different ionic conditions. Overall, the figure is hard to understand, especially panels d-i are overloaded with electron density in mesh representation. I suggest that the authors improve this figure by cleaning it up, which should help understanding their interpretations. For

example, panels e, g, i could only show the difference maps. Especially in e, substantial densities are visible in the difference map what appears to be behind the filter, similar densities albeit less pronounced are found in g and i. This may point to other effects Ca²⁺ has on NavAb structure and function. More information and explanation of this observation is critically needed.

I'm very sorry to bother you with the complicated Fig3. We apologize again for this complication and for causing unnecessary concerns. Since the crystals of these channels have four-fold symmetry, the electron density observed in the depth direction is the same as that of the left or right subunits rotated 90 degrees. Therefore, the depth direction information was unnecessary, only to obscure the figure. As you pointed out, we have made Fig3 only a difference map. The raw electron density was SupFig7. Related to the concern below, we added the distance information to understand the interaction. In addition, the four-fold axis of symmetry was added to Fig.3.

- Line 144 and 151: The authors describe interactions of the "carboxyl group" of the Q176 side chain. Glutamine does not have a carboxyl side chain! The electrostatics between Glu and Gln are different, since Gln (Q) is formally uncharged. Interaction with Ca²⁺ can only occur through lone ion pairs, not attractive forces between positive and negative charges.

Excuse me. we ended up typoing in a critical part. These carboxyls are carbonyls.

- For all interactions described, distances need to be given in the figures. At this point it is not possible for the reader to judge the described interactions.

Fig 3 has been simplified, and distance information has been added. In the horizontal view, we provide the distance information from ions or water in Site_{HFS} and Site_{CEN} to protein atoms. In a vertical view, we added the distance information from the atoms of proteins to ions or waters in Site_{IN}. In the horizontal view, we provide the distance information from ions or water in Site_{HFS} and Site_{CEN} to protein atoms. In a vertical view, we added the distance information from the atoms of proteins to ions or waters in Site_{IN}.

- Figure legend: line 618: "horizontal and vertical view of the ion pathway" – which one is it?

In the new figure, it will be Fig3d-h. Scales are listed at the side of the diagram for distance information. Also, in the previous version, there was an error in the scale, which has been corrected. It was also shown in the figure that it is symmetrical four times.

- In the last paragraph about the structure, the authors mention that it is unlikely for 4 Ca²⁺ ions to bind. The occupancies obtained from their x-ray structures need to be reported to make a case here. Again, in the current state the reader cannot evaluate the claims.

Occupancy and temperature factors for ions and water molecules around the selective filters have been added to SupTable2. As a result, we can conduct a detailed analysis in response to your suggestions, which enabled us to have specific discussions. Thank you for your suggestions.

As you can see from this table, there is enough occupancy for the Ca ions in the selective filter, but the temperature factor is higher than that of the surrounding protein atoms. This may be due to the fact that the atom has an atomic number higher than the actual atom or that this ion can move freely.

These causes suggest that calcium ions are placed where water and calcium ions are mixed and that water and calcium ions can move freely to some extent. However, the resolution was insufficient, and the crystal structure alone made it impossible to determine the exact number of calcium ions in the selectivity filter. From this point of view, we think it has become necessary to evaluate molecular dynamics simulations.

In the previous manuscript, the number of calcium ions was determined in this section of crystal structure. Still, it should have been determined based on the results of molecular dynamics simulations. In the crystal structure section, we mentioned that it is difficult to determine the number of calcium ions with this resolution. In the MD simulation section, we clarified that only one calcium ion exists.

- I am not an expert in MD simulations. However, the data representation in Fig 4 and 5 should be improved and more detailed description should be added. Better explanation of the simulations and how energies were extracted will help the general reader to follow the line of argumentation. A structural representation along the z-axis (basically showing the ion conduction pathway) could make it easier to understand the details.

To make the permeation process easier to understand, we created movies corresponding to trajectories such as Fig4, SupFig7, and SupFig8. By the movies, we hope it makes it easier to understand the following things. Significantly when ECC is not applied, Ca inhibits the current even in the wild-type channel. In the wild type under ECC application, Na permeation occurs in the presence of Ca, and in mutants, Ca inhibits the Na permeation. For Fig5, an axis explaining the positions of r has been added to Fig5b. In addition, a schematic diagram for calculating one-dimensional free energy has been added to SupFig11.

- Lines 279-281: "The blocking mechanism differs between mutations to hydrophilic residues and mutations to smaller residues (Fig.6e and f)." This is an important limitation that the authors point out themselves and contradicts the original assumption and the title. It directly shows that there is not a universal mechanism that explains Ca²⁺ block of Na⁺ channels. This also limits the approach and the model-nature of NavAb. The authors should be more careful in their generalization. This loops back to my earlier comment on homology between NavAb and NMDAR or AMPAR.

We are very appreciative of your scrutinizing in detail. The part you pointed out was very short of words.

Also, we wanted to show that mutations of apparently different properties can produce similar results. This result could not be clarified by mutational analysis only. Therefore, we gave a careful explanation to ensure that this claim was conveyed correctly.

So we rewrote the first sentence of the Conclusion as follows:

Electrophysiology experiments indicated that divalent cation blocking is caused by two types of mutations: hydrophilic and smaller-side-chain mutations. The property of these two types of mutations is apparently different, and it was surprising that the same inhibitory effect appeared despite such differences. The crystal structures of these mutants show an increment of the electron density derived from calcium ions at the bottom of the selective filter (Fig.3). In the small-side-chain mutant, electron densities of additional water molecules were found in the space created by the smaller-side-chain mutation. The molecular dynamics simulation indicated one calcium ion in the pathway of the divalent cation-blocking mutants. These results suggested that the mutations of different properties also provide similar inhibitory mechanisms (Fig. 7). Hydrophilic side-chain mutations provide hydrogen bonds to molecules around the Site_{IN} of the selective filter, allowing calcium ions to stack at Site_{CEN}, the center of the ion permeation pathway (Fig. 7:middle). Small-side-chain mutation of Leu176 creates extra cavities where the original leucine side chains were. The inclusion of additional water molecules here allows divalent cations to interact with water molecules and block the entrance to the inner vestibule (Fig. 7:right), similar to hydrophilic mutations on the residues.

As mentioned above, we mentioned that generalization to NMDAR still has challenges in the final paragraph of the Conclusion.

Finally, we think that we improved the quality of our manuscript, such as organizing the points of discussion, by receiving many comments by the reviewer. I therefore appreciate your comments again, and we hope we have dispelled your concerns.

Reviewers' Comments:

Reviewer #1:

Remarks to the Author:

The authors have satisfactorily addressed my concerns

Reviewer #2:

Remarks to the Author:

The revised version of the manuscript is considerably improved and concerns are adequately addressed.

As it is now, the manuscript will be interesting and helpful for researchers in the field.